# Eco-Friendly Electroless Template Synthesis of Cu-Based Composite Track-Etched Membranes for Sorption Removal of Lead(II) Ions

**DOI:** 10.3390/membranes13050495

**Published:** 2023-05-07

**Authors:** Liliya Sh. Altynbaeva, Anastassiya A. Mashentseva, Nurgulim A. Aimanova, Dmitriy A. Zheltov, Dmitriy I. Shlimas, Dinara T. Nurpeisova, Murat Barsbay, Fatima U. Abuova, Maxim V. Zdorovets

**Affiliations:** 1The Institute of Nuclear Physics of the Republic of Kazakhstan, Almaty 050032, Kazakhstan; lilija310378@gmail.com (L.S.A.); d.shlimas@inp.kz (D.I.S.); mzdorovets@inp.kz (M.V.Z.); 2Department of Chemistry, L.N. Gumilyov Eurasian National University, Astana 010008, Kazakhstan; 3Department of Nuclear Physics, New Materials and Technologies, L.N. Gumilyov Eurasian National University, Astana 010008, Kazakhstan; 4Department of Chemistry, Hacettepe University, Ankara 06800, Turkey; mbarsbay@hacettepe.edu.tr; 5Department of Intelligent Information Technologies, The Ural Federal University, Yekaterinburg 620002, Russia; 6Engineering Profile Laboratory, L.N. Gumilyov Eurasian National University, Astana 010008, Kazakhstan

**Keywords:** electroless deposition, copper microtubules, track-etched membranes, template synthesis, lead(II) ions removal, sorption capacity

## Abstract

This paper reports the synthesis of composite track-etched membranes (TeMs) modified with electrolessly deposited copper microtubules using copper deposition baths based on environmentally friendly and non-toxic reducing agents (ascorbic acid (Asc), glyoxylic acid (Gly), and dimethylamine borane (DMAB)), and comparative testing of their lead(II) ion removal capacity via batch adsorption experiments. The structure and composition of the composites were investigated by X-ray diffraction technique and scanning electron and atomic force microscopies. The optimal conditions for copper electroless plating were determined. The adsorption kinetics followed a pseudo-second-order kinetic model, which indicates that adsorption is controlled by the chemisorption process. A comparative study was conducted on the applicability of the Langmuir, Freundlich, and Dubinin–Radushkevich adsorption models to define the equilibrium isotherms and the isotherm constants for the prepared composite TeMs. Based on the regression coefficients R^2^, it has been shown that the Freundlich model better describes the experimental data of the composite TeMs on the adsorption of lead(II) ions.

## 1. Introduction

With the significant increase in social awareness and current concerns about recycling and environmental issues, scientists have focused particularly on solving the problem of effective and at the same time gentle wastewater treatment [1,2]. Heavy metal ions (e.g., arsenic, lead, mercury, copper, cadmium, and chromium), do not degrade compared to some common organic pollutants and can accumulate in living organisms and plants, causing serious damage to the environment and human health due to their toxicity and carcinogenicity [3,4,5,6]. Researchers have paid particular attention to the disposal of lead ions, as lead–acid batteries have been the dominant device in large-scale energy storage systems since 1859 [7,8]. The high neurotoxicity and haematotoxicity of lead(II) ions, their marked toxic effects on major human organs, as well as their ability to accumulate in the body for a long time, are the main reasons for the high effort in developing materials and methods for their removal [9,10]. Various methods such as sedimentation, oxidation, electroless deposition, electrochemical reduction, ion exchange, biosorption, and adsorption are widely used to remove toxic metal ions from wastewater [11,12]. However, each method has certain limitations in its application and is usually restricted by technical and economic problems. Furthermore, these methods have a number of disadvantages such as incomplete removal, high energy costs, formation of toxic sludge or waste that need to be disposed of, and they are uneconomical for the removal of heavy metals at lower concentrations. The above-mentioned disadvantages of these water treatment methods emerge as an advantage for the adsorption process due to their ease of application, high efficiency, low cost, etc. [13,14]. A significant number of various lead ion adsorbents have been reported in the literature, including those based on nanomaterials and nanocomposites [15,16,17,18]. However, many of the existing sorbents need to be improved in terms of production costs, time to reach sorption equilibrium, and adsorption efficiency. In this context, it is very important to develop new types of efficient adsorbents that provide high lead(II) removal in a short time.

Composite track-etched membranes (CTeMs) are ordered arrays of microtubes or nanowires deposited by chemical or electrochemical templating synthesis into polymer templates based on track-etched membranes (TeMs) [19]. This class of Cu microtube-based functional composites has already demonstrated its undoubted potential in wastewater treatment, both as highly efficient catalysts [20,21,22] and as sorbents for various pollutants such as arsenic (III) ions [23,24,25]. TeMs with embedded metallic microtubes (MTs) have many practical advantages: a wide range of desired metals/alloys can be easily deposited within the channels of a membrane; the membrane geometry can be prepared with desired parameters and allow the active metallic phase to be uniformly distributed over the whole porous surface; it is possible to carry out any reaction or experiment without removing the porous catalysts or sorbents from reaction media after the reaction is complete. The proposed CTeMs can be successfully used in cross-flow [24] and bath [23] modes as an adsorbent for the removal of As(III) from wastewater samples.

The typical plating solution for copper electroless deposition involves the use of formaldehyde (CHOH) as a reducing agent, but this approach is becoming less popular due to formaldehyde’s high toxicity and trace amounts found in the final composites [26]. Formulations based on non-toxic reducing agents such as ascorbic acid (Asc) and glyoxylic acid (Gly), dimethylamine borane (DMAB), etc., are particularly noteworthy as an alternative to formaldehyde-based plating solutions [27].

In this research, Cu microtube-based CTeMs samples were synthesized using the above-mentioned environmentally friendly reducing agents. Optimal template synthesis conditions were determined for Gly- and DMAB-based composites. Similar studies on the synthesis of CTeMs using CHOH and Asc-based plating solutions have been published in our previous works [28,29]. This study also includes a comparison of the effect of plating solution composition on the Pb (II) sorption properties of the composites, as well as the mechanism of sorption.

## 2. Materials and Methods

### 2.1. Materials

Copper sulfate pentahydrate (CuSO_4_·5H_2_O), tin(II) chloride (SnCl_2_), potassium sodium tartrate (KNaC_4_H_4_O_6_ × 4H_2_O), palladium chloride (PdCl_2_), formaldehyde (CHOH), ascorbic acid (Asc), glyoxylic acid (Gly), sodium lauryl sulfate, ethylenediaminetetraacetic acid (EDTA), and dimethylamine borane (DMAB) were all purchased from Sigma-Aldrich (Schnelldorf, Germany) and used without further purification. The certified reference solution of 0.1 g/L Pb(II) was purchased from Ecroskhim (Sankt-Petersburg, Russia). The water used in all the experiments was purified using a D-301 water purification system (Akvilon, Russia) with a resistivity of 18.2 MΩ/cm.

### 2.2. Composite TeMs Electroless Synthesis

The polymer template used in all experiments was PET ion-track membranes with a pore density of 4 × 10^7^ cm^−2^ and an initial pore diameter of 435 ± 10 nm. The pristine polymer film was irradiated using the DC-60 accelerator of heavy ions (accelerated ion—^15^Kr^84^, energy of 1.75 MeV/nucleon, the film thickness is 12.0 microns) and successively etched in NaOH (2.2 M, 85 °C). Sensitization was carried out by rinsing samples in a solution containing 50 g/L SnCl_2_ and 60 ml/L HCl (37%) for 6 min and washing thoroughly with hot water for 2–3 min. The sensitized PET TeMs sample was then activated by immersion in a solution of 0.1 g/L PdCl_2_ and 10 ml/L HCl (37%) for 6 min [29].

Detailed information about the composition of copper plating solutions and experimental conditions are given in Table 1. After the deposition was completed, the composite was washed in deionized water and dried in an oven at 60 °C for 20 min until constant mass.

### 2.3. Characterization of PET Template and Composites

The composites produced were characterized by various techniques. Scanning electron microscopy (SEM) images were taken using a JEOL JFC-7500F microscope (Tokyo, Japan). Prior to the SEM analysis, a 15 nm gold layer was sputtered onto the membranes. To obtain high-quality cross-sectional images of the composite films, the composite membrane was irradiated with a UV lamp for 10 days from each side, then broken, and the images were taken. The elemental composition of the composites was investigated using a Hitachi TM3030 SEM (Hitachi Ltd., Chiyoda, Tokyo, Japan) equipped with a Bruker XFlash MIN SVE (Bruker, Karlsruhe, Germany) microanalysis system at an accelerating voltage of 15 kV.

The pore size of the original template and the structural parameters of the copper MTs were determined by the porometry method using the Hagen–Poiseuille Equation (1) [33]:(1)$$Q=\frac{8\pi }{3MRT}\sqrt{\frac{nr^{3}\Delta p}{l}}$$
where Δ*p* is the pressure difference, MPa; *M* is the molecular mass of the gas g × mol^−1^; *R* is the universal gas constant, erg/(mol × K); *n* is the number of microtubes per square centimeter of membrane area (template pore density); *l* is the membrane thickness, cm; and *T* is the temperature, K.

X-ray diffraction (XRD) patterns were obtained using a D8 Advance diffractometer (Bruker, Karlsruhe, Germany) to examine the crystalline structure of the samples. The X-ray was generated at 25 mA and 40 kV, and the scanning position ranged from 30° to 90° 2(θ). The average crystallite size was determined using the Scherrer equation [34].

The surface morphology of the composite membranes was studied by a scanning probe microscope (SmartSPM-1000, NT-MDT, Novato, CA, USA) in semicontact mode using an NSG10 (TipsNano, Tallinn, Estonia) rectangular-shaped silicon cantilever (length, 95 ± 5 μm; width, 30 ± 5 μm; thickness, 1.5–2.5 μm; probe tip radius, 10 nm; resonance frequency, 200 kHz). Initial scanning of a 10 × 10 μm^2^ sample was performed at a speed of 5.0 μm/s. Average roughness was calculated from a scan area of 3 × 3 μm^2^. The obtained data were processed and analyzed using IAPRO-3.2.2 software.

The amount of deposited copper was determined gravimetrically based on the difference in the weights of the composite before and after plating with an accuracy of 0.1 mg (AS 220.R2, Radwag, Radom, Poland) and expressed in mg/cm^2^.

The charge on the adsorbent surface was investigated in the pH range of 3.0 to 9.0 and the zero point of charge (pH_zpc_) value was determined as described in [35]. Briefly, 10 mL of NaCl solution (0.01 M) was brought to the desired pH value (pH_i_) by adding 0.1 M HCl_(aq)_ or NaOH_(aq)_. Then, a sample of 2 × 2 cm of composite membrane was added to each flask and shaken on a shaker (IKA KS 3000i, Konigswinter, Germany) for 12 h at room temperature. CTeMs were removed from the solution by filtration and the final pH (pH_f_) of the filtrate was measured using a pH meter (HI2020-02, HANNA Instruments, Smithfield, VA, USA).

### 2.4. Batch Absorption Experiments

All experiments conducted to determine the Pb(II) adsorption performance of composite TeMs and pristine templates were carried out using batch equilibrium techniques. Feed Pb(II) solution (100 ppm, pH 4.0) was prepared by diluting the certified Pb(II) reference solution (0.1 g/L, Ecroskhim, Russia). Disposable plastic vials (Isolab, Eschau, Germany) containing 15.0 mL solution and 2 × 2 cm composite adsorbate were shaken (100 rpm), IKA KS 3000 IS control, (IKA, Konigswinter, Germany) at room temperature for different periods ranging between 15 min and 10 h. Each experiment was repeated in triplicate. The concentration of Pb(II) in the aliquots was determined by ICP–MS (Thermo Fisher Scientific, XSeries 2, Bremen, Germany). The amount of Pb(II) adsorbed was calculated using Equation (2):(2)$$Q_{e}=\frac{\left(C_{0}-C_{e}\right)\times V}{m}$$
where *Q_e_* is the amount of Pb(II) adsorbed by the unit mass of copper (mg/g), *C*_0_ is the feed concentration (mg/L), *C*_e_ is the concentration of Pb(II) in aliquots (mg/L), *V* is the volume of the solution (L), and *m* is the amount of Cu loaded on the membrane (g). In the case where the pristine template was tested, the weight of PET TeM was used in *m* (g).

The effect of pH on Pb(II) adsorption was studied in the pH range of 3 to 10. Other parameters were kept constant (initial Pb(II) concentration: 50 ppm; adsorbent dose: 2 × 2 cm^2^; contact time: 420 min). The pH of the solution was adjusted dropwise with 1.0 M HCl_(aq)_ and 1.0 M NaOH_(aq)_. pH was measured using a digital pH meter, HANNA HI2020-02 (HANNA Instruments, Smithfield, VA, USA). All experiments were performed in triplicate.

## 3. Results and Discussions

### 3.1. Eco-Friendly Template Synthesis of Copper Microtubules (MTs)

#### 3.1.1. Glyoxylic Acid

Glyoxylic acid has the potential to be the best alternative to formaldehyde due to its similar nature and properties [30,36]. However, unlike formaldehyde, the glyoxalate ion in solution has no vapor pressure, and the problem of atmospheric formaldehyde contamination and the corresponding need for control equipment is eliminated. The coating speed and bath stability are superior to a formaldehyde bath under standard conditions. The concentration of EDTA does not affect the morphology of the deposited copper film, and the flexibility of the film is good. The uniformity of the coating of the entire wall surpasses the formaldehyde bath. The working solution has no vapor pressure and showed good reducing ability during the electroless deposition of copper. Therefore, glyoxylic acid can replace formaldehyde and eliminate health and environmental problems. In this study, the effect of copper deposition time and pH were studied. Based on the previous results, pH values of 12.65 and 13.5 were selected for the electroless deposition of copper MTs using glyoxylic acid [21]. Due to the low stability and high temperature of the plating bath (the plating solution becomes unstable in 2–2.5 min after adding the reducing agent), the deposition time ranged from 15 to 60 s. As can be seen from the SEM images presented in Figure 1, the increase in deposition time causes more Cu nanoparticles to accumulate on the surface and inside the nanochannels at both pH values studied. After 60 s of deposition at pH 13.5, the presence of narrowed open nanochannels and a high amount of Cu nanoparticles deposition on the surface is observed (Figure 1g). On the other hand, deposition for 120 s and longer resulted in completely closed membrane pores as seen in Figure 1h.

The deposition rate R of the electroless Cu plating was expressed as the weight gain per 1 cm^2^ of PET TeMs per unit time during the deposition process (Table 1). The solution pH plays a critical role in copper plating and the deposition rate at pH 13.5 was increased 15 times compared to samples prepared at pH 12.65. It is well known that the plating solution should ensure the formation of a uniform and strong deposited metal layer with excellent crystal structure [19,37]. According to the X-ray diffraction data presented in Table 2 (corresponding XRD patterns are shown in Appendix A of the Appendix A), the unit cell of all synthesized Cu_Gly@PET composites deposited at pH 12.65 is characterized by a cubic syngony (Fm3m) with a cell parameter of 3.607–3.594, with the exception of the amorphous sample deposited at pH 12.65 in 15 s. Samples deposited within 60 s at pH 13.5 have a cell parameter close to the reference value (a = 3.6150, PDF #040836) [23]. When the lines on the diffractogram were approximated with the required number of symmetric pseudo-Voigt functions, the widths of the registered lines at half of their height (FWHM) were measured, allowing us to estimate the degree of perfection of the crystal structure and the degree of crystallinity (DC). For all samples prepared at pH 13.5, the DC values were calculated to be higher compared to samples deposited at pH 12.65. Based on the SEM and XRD data, the plating time of 60 s and the plating solution pH of 13.5 were determined as optimal experimental conditions.

#### 3.1.2. DMAB

According to the literature survey, boron-containing compounds, in particular, DMAB, are often used in electroless deposition processes [32,38]. The boron atom in DMAB is bonded to nitrogen by a donor-acceptor bond via the unshared electron pair of nitrogen, in addition to its three covalent bonds with hydrogen. Of the wide range of borazanes, only a few compounds such as dimethylamine borane, diethylamine borane, and pyridinamine borane have found practical application in the manufacture of metal coatings. Monomethylborazane, monoethylborazane, and isopropylborazane are too reactive reduction agents, and their resulting plating solutions are unstable [39]. Trimethylborazane and triethylborazane are weakly reactive, and compounds with longer carbon chains are soluble only in organic solvents, making their use in the reduction of metal ions from aqueous solutions more complicated. The main advantage of DMAB is that it offers high stability compared to sodium borohydride reduction and allows the deposition of metallic nanostructures under milder conditions such as lower temperatures (50–70 °C) and over a wide pH range (5–10) [40]. In this study, the effect of deposition temperature on the structure of Cu_DMAB@PET composites after 15 min of deposition was evaluated in the temperature range of 25–55 °C. As can be seen from the SEM images in Figure 2, all temperature modes ensured uniform coating of the PET template.

XRD was used to study the crystal structure of the obtained composite membranes. The XRD pattern of Cu_DMAB@PET (Appendix A) identified three diffraction peaks at 2θ equal to 53.94, 63.12, and 71.72, characteristic of the monoclinic structure of copper(I) oxide and attributed to planes (211), (220), and (310), respectively (JCPDS: 01-073-6023, tenorite). The characteristic diffraction peaks of the Cu phases at 2θ = 43.50 (111), 50.79 (200), 74.63 (220), and 90.27 (311) were also identified in all XRD spectra. The identified planes are consistent with the JCPDS card of Cu (PDF #040836), which indicates a cubic syngony (Fm3m) with the cell parameter of *a* = 3.61, similar to the reference value of the *a* parameter.

Detailed data on the crystal structure of the synthesized composite TeMs are given in Table 3. According to the data obtained, the wall thickness of the copper MTs ranged from 11.5 nm (25 °C) up to 33.8 nm (55 °C). The deposition rate increased linearly with the increasing deposition temperature. Increasing the amount of deposited copper phase, which plays an active role in the sorption process, increases the removal of Pb(II) ions from the aqueous solution. Therefore, a higher MT wall thickness is desirable for a sorbent with enhanced efficiency. As can be seen in Table 3, the crystallinity of the deposited phase is also increased at 55 °C, where the deposition in the nanochannels was the highest. The degree of crystallinity is one of the key parameters that affects the performance of copper-loaded composite TeMs, and samples with high DC value demonstrate higher catalytic activity in the removal of various pollutants [24]. Therefore, Cu_DMAB@PET samples with higher copper MTs wall thickness and crystallinity, deposited at 55 °C, and consisting of Cu_2_O (37.4%) and Cu (62.6%) phases, were studied in further experiments on the sorption of lead(II) ions.

#### 3.1.3. Ascorbic Acid and Formaldehyde

Ascorbic acid (AA) is a mild reducing agent and is most commonly used for the synthesis of nanoparticles of metals such as copper, silver, gold, etc. [41,42,43]. When AA is used as the reducing agent, copper reduction is effective even at low concentrations. Thus, in the first minutes of the reaction, a complex of copper with ascorbic acid is formed [28], which undergoes oxidation–reduction decomposition, forming ultra-dispersed copper and ascorbic acid oxidation products. AA provides a good stabilizing effect by protecting the copper nanoparticles from oxidation and agglomeration during the long deposition period. Moreover, the low toxicity of AA makes it a promising reducing agent for green chemistry.

In a previous study, K. Valenzuela et al. demonstrated the use of ascorbic acid as a copper reducing agent in a template based on microtubules produced using microtubule-associated proteins [31]. However, despite the stabilizer additives added to the plating solution, the solution itself appeared to be unstable and the deposition experiments at pH = 4.0 did not exceed 5 min, resulting in the formation of nanowires with a diameter of about 15 nm. We have previously established that the most efficient copper deposition in the PET template occurs at pH 4.0 and a deposition time of 120 min [21,28]. SEM image of Cu_Asc@PET composite surface is shown in Figure 3a. It is obvious that the membrane surface and pore walls are uniformly covered by copper nanoparticles. The copper MT wall thickness was not more than 13.1 ± 2.0 nm, and the inner pore diameter was calculated as 367.1 ± 3.9 nm according to Equation (1).

The electroless plating of copper MTs in a PET template using formaldehyde directly depends on factors such as temperature [29], proportions of plating solution components [44], conditions of polymer template modification [22], etc. Based on previous studies, we synthesized Cu_CHOH@PET samples with a wall thickness of 44.8 ± 6 nm and internal pore diameter of 320 ± 8 nm, as can be seen in Figure 3b.

According to the detailed X-ray diffraction data (Appendix A), the unit cell in both Cu_CHOH@PET and Cu_Asc@PET composite membranes can be characterized by a cubic syngony (Fm3m) with cell parameters (a) of 3.602 and 3.608, respectively. DC values were calculated as 43.8% for Cu_CHOH@PET and 47.0% for Cu_Asc@PET composite membrane (Appendix A). The structural parameters of Cu_CHOH@PET and Cu_Asc@PET composite membranes are presented in the supporting information (Appendix A). It should be noted, in both types of composites only single copper phase was identified.

### 3.2. Pb(II) Sorption by Composite TeMs

Adsorption is a time-dependent process influenced by the physical/chemical characteristics of the sorbents and sorption conditions. The effect of pH on the adsorption of Pb(II) ions by the deposited nanoparticles in the pH range from 3.0 to 8.0 is shown in Figure 4a. Due to the destruction of the PET template in strongly basic solutions above pH 9.0, the upper limit in the pH range was determined to be 8.

As the initial pH increases from 3.0 to 5.0, the removal efficiency of lead ions increases. At low pH, Pb(II) ion removal was inhibited, probably as a result of competition between H^+^ and Pb(II) ions for available adsorption sites, with an apparent preponderance towards H^+^ ion uptake [45,46]. However, as the pH increased, the negative charge density on the surface of adsorbents increased due to deprotonation of the binding sites, and thus the adsorption of Pb(II) ions increased. This indicates that ion exchange is one of the main adsorption processes. When the pH was in the range of 5 to 7, due to the increase in OH- ions concentration, the Pb(II) ions in the aqueous solution started to convert into Pb(OH)_2_, hindering the adsorption of Pb(II). Therefore, the adsorption capacity gradually decreased when the initial solution pH was higher than 5. Several previous studies have shown that at pH > 6.0, the absorption of Pb(II) ions is significantly affected due to Pb(OH)_2_ formation, and at pH 9.0, Pb(II) ions are almost completely converted to hydroxides in solution [47,48]. It is therefore more appropriate to investigate the sorption tests at a pH lower than 6.0. pH of the point of zero charge (pH_PZC_) defines the pH of the solution at which the total surface charge of the adsorbent is zero. Studying pH_PZC_ is important for identifying the adsorption mechanism and explaining the nature of interactions between the sorbent and the adsorbate [49]. A diagram of ΔpH_final_ versus pH_initial_ is shown in Figure 4b. This figure shows that the pH_PZC_ obtained for all CTeMs types studied was approximately 7.0. Thus, below the respective pH values of 7.0, the surface charge on all studied adsorbents considered was positive, and hence the uptake of Pb(II) ions was low. As seen in Figure 4a, the highest Pb(II) sorption was observed at pH 5 in all composite membranes, as a combinatorial result of the effects discussed above that have positive or negative consequences on absorption. Therefore, further studies were carried out at this optimum pH value.

The variation in the sorption capacity of copper microtube-based composites as a function of time is shown in Figure 4c. For the Cu_DMAB@PET membranes, saturation occurred after 360 min of sorption, while for all other samples (Cu_CHOH@PET, Cu_Asc@PET and Cu_Gly@PET) it was observed after around 480 min. Based on the experimental data obtained, the equilibrium sorption capacity q_e_ of Pb(II) sorption from 50 ppm solution for each composite membrane was determined (Figure 4d). The composite membrane obtained using ascorbic acid as the reducing agent was found to have the highest efficiency for lead ions sorption. The q_e_ value for this composite type is more than 40% compared to membranes obtained using DMAB. The decreasing activity of the Cu_DMAB@PET composites is caused by the presence of Cu_2_O phase, as was demonstrated in our previous work [21].

Atomic force microscopy (AFM) is a unique technique for monitoring the surface topography of materials with great precision. AFM is often used for the characterization of porous TeMs and functional materials based on it [50,51,52]. In this study, we acquired AFM images of the composite membrane surfaces at a scan size of 3 × 3 μm. Figure 5 presents the three-dimensional AFM images and the roughness (R_a_) values calculated for each image. Average roughness values were calculated for at least 10 images with 512 × 512 points taken from different locations. As previously reported, R_a_ increases the contact surface area of sorbents [23], therefore, a rougher surface has a greater effective surface area than a smooth surface, and as a result, stronger interactions can occur on rough surfaces [53]. The roughness of composites obtained using Asc and Gly as reducing agents was found to be around 40% higher than when using DMAB or CHOH-based plating solutions. Therefore, Cu_Gly@PET and Cu_Asc@PET composites have a higher R_a_ due to tiny depressions and convexities and thus offer a larger area for interaction with the lead(II) ions. In addition, the Cu_Asc@PET composite has a smaller crystallite size (19.0 ± 3.5 nm) compared to Cu_Gly@PET (24.9 ± 6.7 nm), we therefore suggest that smaller copper crystallites also contribute to improved adsorption performance, along with increased surface areas, in particularly for the membranes obtained using ascorbic acid as a reducing agent of plating solution.

### 3.3. Study of the Sorption Kinetics

Studying the sorption kinetics makes it possible to determine the factors that affect the dynamics and limit the process rate. The sorption rate is a significant factor that largely determines the possibility of using the studied material as a sorbent. As a general rule, the interaction of a sorbate with a sorbent has a high rate in the first moments of contact, after which it reaches a constant level. In a technological or laboratory process, it is desirable to achieve sorption equilibrium quickly to minimize the sorption cycle time [54,55].

Parameters such as time to reach equilibrium, sorption capacity, selectivity of sorbate extraction, acid-base and complexing properties are essential characteristics of any absorber and are of both practical and theoretical significance. From the integral kinetic dependence of the sorption, it is possible to determine the time required to establish equilibrium in the “sorbent–metal salt solution” system [56]. This characteristic makes it possible to evaluate the possibility and expediency of using the sorbent for solving practical problems. Theoretical processing of kinetic curves using diffusion and chemical kinetics models allows us to finalize the sorption mechanism to identify the limiting stages of the process, which can also be used to solve several practical issues.

For porous sorbents, the adsorption rate can be controlled by external film mass transport (film diffusion) or mass transfer of solutes within the particle (diffusion in the pores or migration along the pore surface), i.e., internal or intra-particle diffusion. Morris–Weber intraparticle diffusion model, pseudo-first-order, pseudo-second-order, and Elovich models were applied for modeling the sorption kinetics of lead(II) ions on copper-based composite TeMs. Figure 6a–d show the kinetic plots for the models studied. The correlation coefficients (R^2^), linearized equations, and parameters calculated from these models are summarized in Table 4.

The Morris–Weber intra-particle diffusion model can be applied when the rate-determining step is based on the mass transfer of the adsorbate to solid surface areas [57]. The linear form of the equation of this model is presented in Table 4. It contains two constants: k_i_ ((mg/g∙min^1/2^)), the rate constant of intra-particle diffusion, and the constant C_i_ (mg/g), representing the resistance to mass transfer due to the boundary layer. According to this model, intra-particle diffusion is the limiting stage if the graph is linear and passes through the origin. Thus, diffusion within particles is the only mechanism controlling the sorption process. Where intra-particle diffusion is not the only limiting stage, the graphs have a multilinear character. As can be noticed from Figure 6a, the plots are linear but not do not pass through the origin, which may be due to the difference in mass transfer rate during the initial and final stages of adsorption. It also appears that there is a boundary of layer resistance at the beginning and intra-particle diffusion is not the sole rate-controlling step, but other kinetic models may simultaneously control the adsorption rate [58].

The pseudo-first-order model proposed by Lagergren as the earliest adsorption kinetic model is used to describe the adsorption behavior of solid adsorbents in liquid media. The values of $k_{1}$ and $q_{e}$ and the coefficient of determination, R^2^, were determined by the linear graph dependency in Figure 6b and listed in Table 4. For the pseudo-first-order kinetic model, a lower value of R^2^ indicates that the adsorption kinetics do not match the pseudo-first-order reaction kinetic model. The first-order kinetic model can be applied to the initial stage of the adsorption process when the concentration of adsorbate ions reaching the adsorbent functional groups is extremely low compared to the active centers. In this case, the number of active centers involved in the adsorption changes slightly over time, and the adsorption process can be reduced mathematically by including the concentration of the functional groups of the adsorbent in the reaction rate constant. In the latter stage of the adsorption process, the adsorption rate is affected by the concentration of the two components, so the order of the adsorption becomes two [23]. If the process is defined by the pseudo-second-order model, the interaction between the adsorbate and the functional group of the adsorbent is strictly stoichiometric; that is, a metal ion occupies an adsorption site [59]. The correlation coefficients determined for the pseudo-second-order kinetics (R^2^ = 0.99) were higher than those calculated for the other kinetic models, indicating that the adsorption kinetic data followed the pseudo-second-order kinetic model. Table 4 shows that the experimental values of adsorption capacity (Qe) are very close to the theoretical adsorption capacity (qe) of the pseudo-second-order kinetic model. The linear dependence in Figure 6c over the entire time interval is obvious. The applicability of the pseudo-second-order kinetic model to all studied composite sorbents leads to the conclusion that chemisorption is the rate-determining step of the process, and the effect of the diffusion stage is negligible. Our findings are in agreement with previously reported data on the adsorption mechanism of Pb(II) by other composite sorbents such as multi-walled carbon nanotubes [60], magnetic NiO@biochar [61], FeS@biochar [62] composites, and CuO nanoparticles synthesized by sputtering method [63].

The Elovich model considers the adsorption process on an energetically inhomogeneous surface and how sorption and desorption affect the solute uptake kinetics. It should be noted that desorption processes have a significant effect when approaching equilibrium. Since the synthesized composite membranes have an inhomogeneous porous structure, this model can adequately estimate the adsorption process. Figure 6d and Table 4 list the kinetic constants obtained from the Elovich equation. The applicability of the Elovich equation for the kinetic data indicates that the Elovich equation can properly describe the kinetics of lead adsorption on Cu_Gly@PET and Cu_CHOH@PET samples with high correlation coefficients (R^2^) for this kinetic model. The higher value of initial adsorption rate (α) for Cu_Asc@PET and Cu_Gly@PET composites may be due to the larger surface area of these adsorbents, as confirmed by the AFM analysis presented above.

### 3.4. Study of the Sorption Mechanism

The sorption of heavy metal ions is a complex process influenced by several factors. Possible mechanisms by which this process proceeds include chemisorption, complexation, and adsorption on the surface and in the pores of the sorbent, accompanied by complexation, ion exchange, microprecipitation, and precipitation of heavy metal hydroxides [64]. The most preferred approach to investigate the mechanism of adsorption is the study of isotherms. Sorption isotherms show the distribution of metal ions between the adsorbent and the liquid phase at equilibrium as a function of concentration. The study of these isotherms allows us to conclude the nature of the sorbent surface and the sorbate–sorbent interactions. Thus, to determine the parameters characterizing the sorption properties of composite track membranes, sorption isotherms of lead ions from aqueous solutions were studied using Langmuir, Friendlich, and Dubinin–Radushkevich models.

According to Langmuir’s adsorption theory, adsorption is localized and occurs at equivalent active sites. Each active site holds only one molecule, and adsorption saturation occurs when the active sites are filled. The adsorbed molecules do not interact with each other and desorb after a certain time; thus, a dynamic equilibrium is established [65,66]. Langmuir isotherm constants b and Q_0_ are related to the features of the adsorbent-adsorbate pair. The Q_0_ and b values were determined from the slope and intersection of the lines on the graph, respectively, in the corresponding coordinates of the linearized isotherm equation (Figure 7) and are presented in Table 5.

The Freundlich model isotherm equation is used to describe adsorption on a heterogeneous surface. Since the adsorption sites in this model have different energy values, the active sorption sites with the maximum energy are the first to be filled. Figure 8 shows the experimental data on the adsorption of lead ions on the composite membranes studied in the coordinates of the linearized Freundlich equation $lnq_{e}=lnk_{F}+\frac{1}{n}lnC_{e}\;$.

The constant *n* is an empirical parameter related to the adsorption intensity, which varies as a function of the heterogeneity of the adsorbent. For optimum adsorption, the values of *n* should be in the range of 1–10, [67]. The $k_{F}$ values for Cu_Asc@PET and Cu_Gly@PET samples were 4.55 and 3.26 mg/g, respectively, which are higher than those for the DMAB and formaldehyde samples, indicating their improved adsorption capacities. This is consistent with the Langmuir isotherm data. For all studied samples the values of Freundlich constant (*n*) are equal to one, indicating the possibility of adsorption of Pb(II) on the composite surface in the first step, the bond energy between the sorbent and lead ions decreases as the surface is filled. The regression coefficients for linear graphs are equal to one, which indicates that the obtained experimental data on adsorption are quite consistent with the Freundlich adsorption isotherm model.

The Dubinin–Radushkevich (DR) isotherm model was developed to account for the effect of the porous structure of the adsorbents [68]. It was based on the adsorption potential theory and assumed that the adsorption process was related to micropore volume filling as opposed to layer-by-layer adsorption on pore walls. It assumes that the adsorption has a multilayered nature in which Van der Waals forces are involved. The DR isotherm model was superior to the Langmuir isotherm since it did not consider a homogeneous surface or constant adsorption potential. DR isotherm model has frequently been used to identify the course of physical or chemical adsorption, as the DR constant β is used to determine the sorption energy (Figure 9). The DR isotherm is a semi-empirical equation and is applied where adsorption follows a pore-filling mechanism. This isotherm is most applicable to physical adsorption processes [69].

The average free energy of adsorption $E_{DR}$, which represents the free energy of the system due to the transfer of one mole of ions from the solution to the solid surface, was calculated from the values of β using Equation (3):(3)$$E_{DR}=\frac{1}{\sqrt{-2\beta }}$$

Determining the value of $E_{DR}$ is important because its numerical value gives an indication of the nature of the interaction forces between lead ions and the active sites on the composite surface. When the adsorption energies $E_{DR}$ are less than 8 kJ/mol, physical adsorption is considered to occur. Adsorption energies between 8 and 16 kJ/mol indicate that the process is driven by the ion exchange mechanism, and chemisorption is observed when $E_{DR}$ values are between 20 and 40 kJ/mol [69]. The adsorption energy values calculated for Cu_CHOH@PET and Cu_Gly@PET composites show the ion-exchange pattern of sorption, while the high $E_{DR}$ values for the Cu_Asc@PET and Cu_DMAB@PET membranes indicate the chemical character of adsorbate–sample interaction. However, it should be noted that the data do not match well with the DR adsorption isotherm model, as the R^2^ values are not close to one. In other studies, for example, using Fe^0^/C ceramsites, the adsorption free energy was calculated as 7.68 kJ/mol, and the interactions between lead(II) ions and the mixed composite were of physical type [70]. Similar levels of interactions (6.294 kJ/mol) have been reported for the adsorption of lead by a chitosan–polyacrylonitrile blend [71]. The sorption energy of the nanocrystalline TiO*_2_* was found to be 2.996 kJ/mol, which also confirms the physical adsorption phenomenon.

A comparison of the adsorption capacities of different Pb(II) adsorbents and copper-loaded composite TeMs attained in this study is presented in Table 6. It should be noted that it is rather difficult to directly compare the data of various studies, as some determining parameters, such as the amount of loaded sorbent, agitation speed, pH, and temperature, are not exactly the same. Overall, the adsorption capacity appeared to be low, but still, given the sustainability of the synthesis method, it could be concluded that composite TeMs, particularly those prepared using ascorbic acid as the reducing agent can be promising and effective eco-friendly alternatives for the removal of lead ions from aqueous media by sorption.

## 4. Conclusions

In this work, a comparative study of chemical template synthesis of Cu microtubules in a PET template was carried out using environmentally friendly non-toxic reducing agents, namely ascorbic acid, glyoxylic acid, and dimethylaminborane. Two plating solutions with pH values of 12.65 and 13.49 were used for copper plating using Gly. It was shown that the specific copper deposition rate increased 15 fold by increasing the pH of the solution. After 45 s of deposition, tubular copper nanostructures with wall thicknesses of 31.6 and 103.9 nm were formed. X-ray diffraction analysis of the crystalline structure and phase composition of the samples showed that when DMAB was used as the copper-reducing agent, two phases were formed, the crystalline copper and Cu_2_O-courite phases. The effect of temperature on the copper plating rate was investigated and the structural parameters of the composites were determined. When Asc is used as a reducing agent, a 10–15 nm monolayer of copper nanoparticles is formed in the template structure.

The sorption removal of lead ions from aqueous solutions by different types of composites (Ag@PDMAEMA-g-PET, Cu_CHOH@PET, Cu_DMAB@PET, Cu_Gly@PET, Cu_Asc@PET) was studied. In all sorption experiments, the pH effect on the removal efficiency of Pb(II) was investigated and the values of the equilibrium sorption capacities of the sorbents were calculated. Experimental adsorption data were analyzed using Langmuir, Freundlich, and DR isotherm models to describe the sorption of Pb(II) by the composite membranes and the inherent mechanisms of the sorption process. The constants and parameters of the adsorption models were determined. By comparing the mean-square deviations (R^2^), it was concluded that the Freundlich model best describes the adsorption of Pb(II) on the surface and in the pores of CTeMs.

The $E_{DR}$ values obtained using the Dubinin–Radushkevich isotherm model suggested that the adsorption process occurs by an ion exchange mechanism when using Cu_CHOH@PET and Cu_Gly@PET composites, but for Cu_Asc@PET and Cu_DMAB@PET membranes, the adsorption process may have a chemical nature.

The adsorption behavior of Pb(II) ions onto the copper microtubes-based composite membranes was studied by the Elovich rate equation, Morris–Weber intra-particle diffusion model, and pseudo-first-order and pseudo-second-order kinetic models. The kinetic adsorption fits the pseudo-second-order model better for all studied composites and Elovich’s rate equation for Cu_Gly@PET and Cu_CHOH@PET samples, indicating the rate-limiting step in the adsorption of Pb(II) ions can be attributed to the chemical interactions between the metal ions and the functional groups at the surface of composite membranes. The adsorption kinetics indicated that the adsorption equilibrium was reached within 480 min under the investigated experimental conditions, and intraparticle diffusion was not the only rate-controlling step in the adsorption mechanism.

Obtained results show that copper plating solutions based on environmentally friendly reducing agents, especially ascorbic acid, can be as effective as, or even more than, conventional formaldehyde reducing agents, as well as provide a greener and more sustainable way to remove toxic lead(II) ions from water.

## Figures and Tables

**Figure 1 membranes-13-00495-f001:**
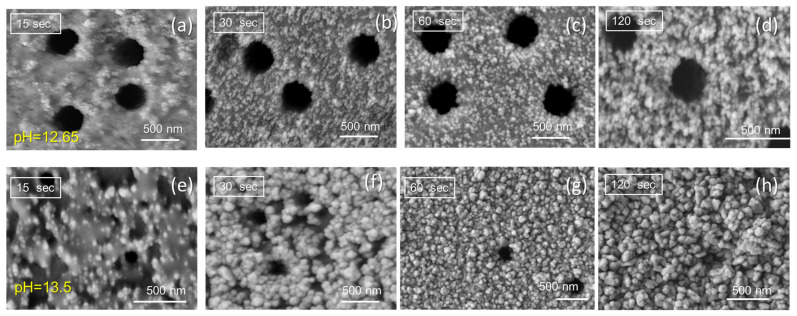
Scanning electron microscopy (SEM) images of Cu_Gly@PET composite membranes deposited at different deposition time and pH values of 12.65 (**a**–**d**) and 13.5 (**e**–**h**).

**Figure 2 membranes-13-00495-f002:**
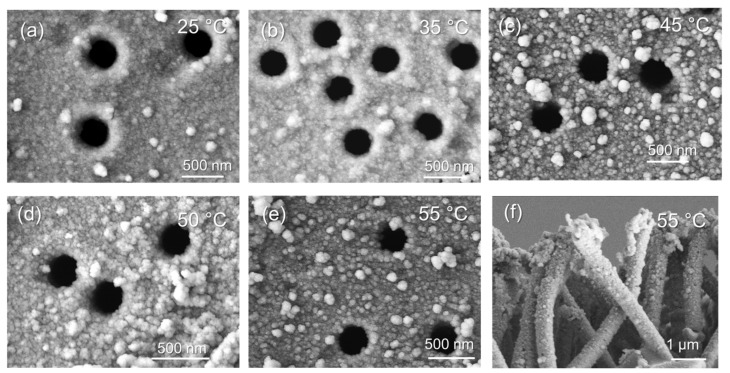
SEM images of Cu_DMAB@PET composite membranes deposited at different temperatures for 15 min (**a**–**e**), and the copper MTs released from the sample deposited at 55 °C (**f**).

**Figure 3 membranes-13-00495-f003:**
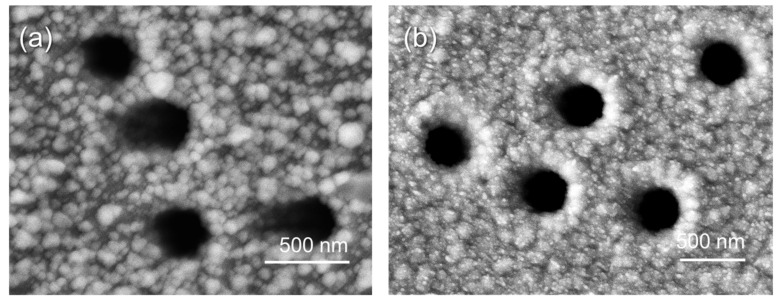
SEM images of Cu_Asc@PET (**a**) and Cu_CHOH@PET (**b**) composite membranes.

**Figure 4 membranes-13-00495-f004:**
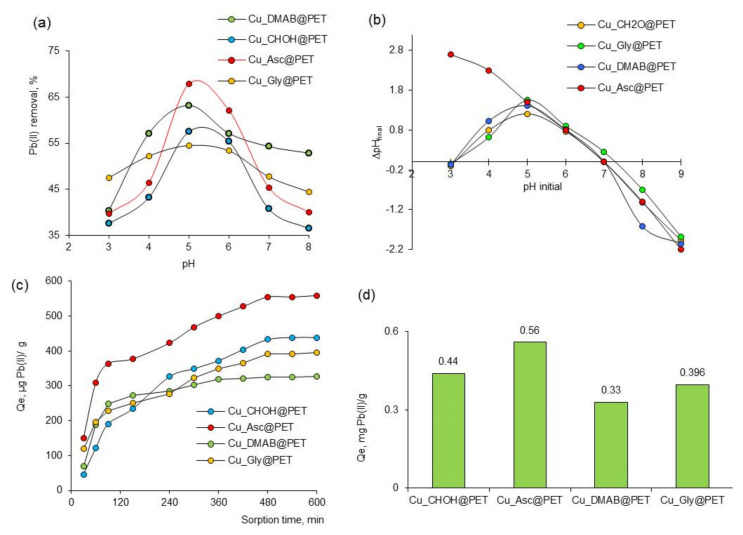
Removal of Pb(II) as a function of solution pH (Pb(II) concentration: 50 ppm; composite TeMs: 2 × 2 cm^2^; contact time: 120 min) (**a**); pH point zero charge (pH_PZC_) plot (**b**); effect of contact time on the sorption of Pb(II) ions (**c**) and equilibrium sorption capacity (**d**).

**Figure 5 membranes-13-00495-f005:**
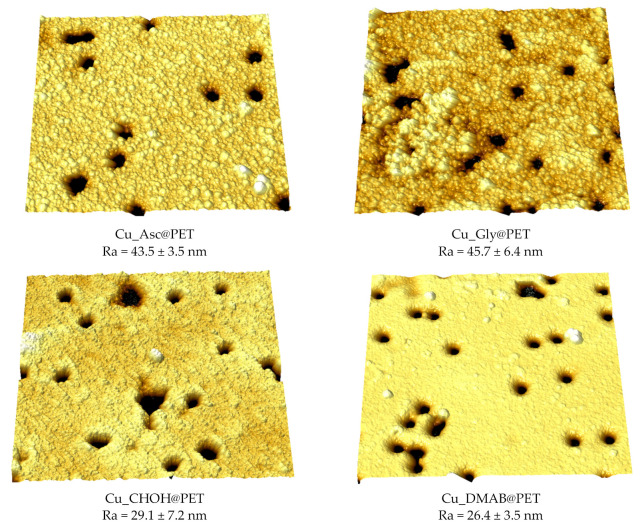
Atomic force microscopy (AFM) images of the surfaces of composite membranes with a scanning area of 3 × 3 µm^2^.

**Figure 6 membranes-13-00495-f006:**
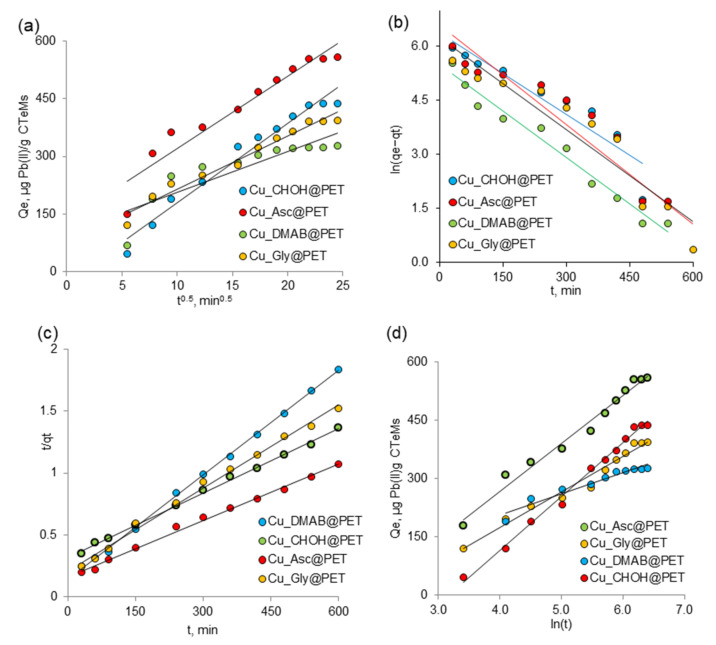
Kinetic model plots of Morris–Weber intra-particle diffusion model (**a**), pseudo-first-order (**b**), pseudo-second-order (**c**) and Elovich model (**d**).

**Figure 7 membranes-13-00495-f007:**
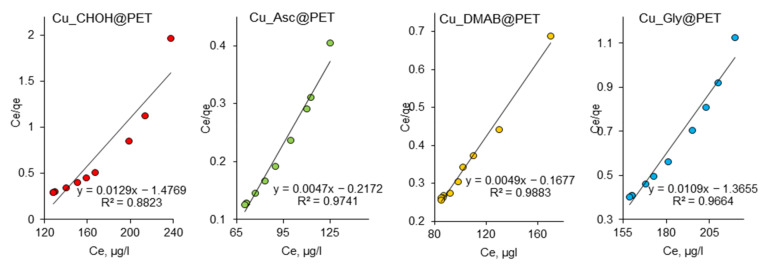
Langmuir adsorption isotherms of Pb(II) adsorption on composite TeMs.

**Figure 8 membranes-13-00495-f008:**
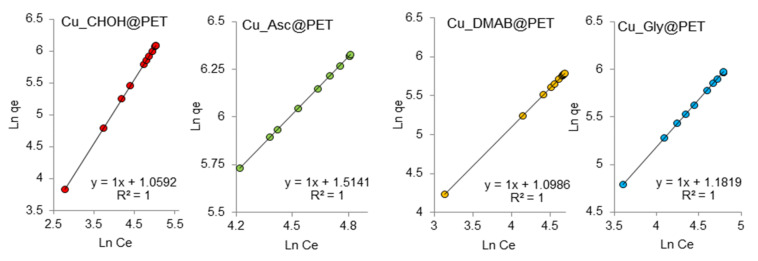
Freundlich adsorption isotherms of Pb(II) adsorption on composite TeMs.

**Figure 9 membranes-13-00495-f009:**
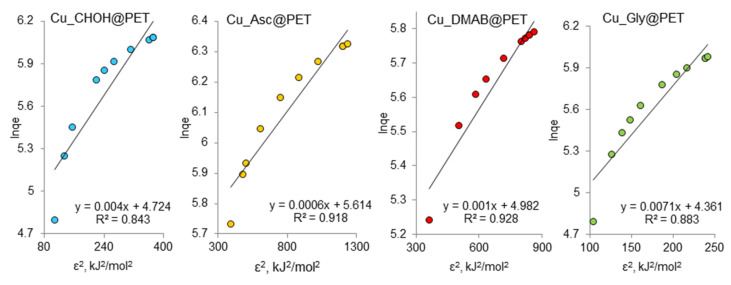
Dubinin–Radushkevich (DR) adsorption isotherms of Pb(II) adsorption on composite TeMs.

**Table 1 membranes-13-00495-t001:** Plating conditions for electroless deposition of copper microtubules (MTs).

Reducing Agent/Sample Code	Plating Bath Composition	Plating Conditions			Ref.
		pH	T, °C	Plating Time, Min	
Glyoxylic acid/ Cu_Gly@PET	CuSO_4_×5H_2_O—7.63 g/L; EDTA—10.26 g/L; Sodium lauryl sulfate—4.0 mg/L; Gly—8.14 g/L	pH = 12.65–13.49 (12.0 M KOH)	70	0.25–0.75	[30]
Ascorbic acid/ Cu_Asc@PET	CuSO_4_ × 5H_2_O—9.6 g/L; CH_3_COOH—10.0 ml/L; Asc—8.2 g/L	pH = 4.0 (9.0 M KOH)	25	120	[31]
Dimethylamine borane/ Cu_DMAB@PET	CuSO_4_ × 5H_2_O—10 g/L; EDTA—14 g/L; DMAB—6 g/L	pH = 1.85 (9.0 M KOH)	26–55	15	[32]
Formaldehyde/ Cu_CHOH@PET	KNaC_4_H_4_O_6_ × 4H_2_O—18 g/L; CuSO_4_ × 5H_2_O—5 g/L; NaOH—7 g/L, CHOH—0.13 M	pH = 12.45 (H_2_SO_4_)	10	40	[23]

**Table 2 membranes-13-00495-t002:** Crystal structure of composite membranes prepared using glyoxylic acid (Cu_Gly@PET) according to XRD data.

pH	Deposition Time, s	Cu MTs Wall Thickness, nm	R, mg/cm^2^ × h	L ^a^, nm	DC ^b^, %	*d* ^c^ Å	Cu Phase Content, %
12.65	15	18.4	1.92	Amorphous			
	30	20.2		13.75	46.9	3.607	100
	60	41.6		24.7 ± 7	47.7	3.605	100
13.5	15	43.3	28.8	17.97 ± 5.1	53.0	3.594	100
	30	84.6		22.61 ± 5.3	57.6	3.603	100
	60	103.9		24.87 ± 6.9	61.7	3.613	100

^a^ average crystallite size, ^b^ degree of crystallinity, ^c^ crystal lattice parameter.

**Table 3 membranes-13-00495-t003:** XRD data of composite membranes prepared using DMAB as reducing agent.

Deposition Temperature, °C	Phase Content	(hkl) ^a^	2θ°	d, Å ^b^	L, nm ^c^	FWHM ^d^	Cell Parameter, Å ^e^	DC, % ^e^	Copper MTs Wall Thickness *l*, nm	Deposition Rate, R, mg/cm^2^ × h
25	Cu_2_O/55.1	211	53.94	1.70	7.76	1.28	4.241	53.0	11.53	0.62
		220	63.12	1.47	162.49	0.06				
		310	71.72	1.31	242.19	0.05				
		111	43.50	2.08	13.51	0.70	3.611			
	Cu/44.9	200	50.79	1.80	35.33	0.28				
		220	74.63	1.27	588.1	0.19				
		311	90.27	1.09	189.51	0.07				
35	Cu_2_O/48.3	211	36.90	2.43	113.44	0.08	4.215	60.4	23.71	0.70
		220	42.95	2.10	45.22	0.21				
		221	53.03	1.73	50.24	0.20				
	Cu/51.8	111	43.38	2.08	126.31	0.08	3.611			
		200	50.41	1.81	145.28	0.07				
		311	90.04	1.09	82.29	0.15				
45	Cu_2_O/46.6	211	53.62	1.71	11.000	0.90	4.183	61.0	27.73	0.71
		310	71.10	1.32	66.41	0.16				
	Cu/53.0	111	43.44	2.08	25.41	0.87	3.608			
		220	74.23	1.28	37.46	0.30				
50	Cu_2_O/42.0	211	53.46	1.71	8.34	1.19	4.1980	56.3	29.39	0.73
		220	62.48	1.49	61.37	0.17				
		221	66.71	1.40	47.78	0.22				
		310	70.97	1.33	62.59	0.17				
	Cu/58.0	111	43.41	2.09	163.85	0.06	3.607			
		200	50.64	1.80	80.41	0.12				
		220	74.59	1.27	148.62	0.06				
		311	90.17	1.09	47.47	0.26				
55	Cu_2_O/37.4	211	53.64	1.71	5.57	1.78	4.205	67.0	33.80	0.78
		220	62.37	1.49	84.55	0.12				
		221	66.61	1.40	56.54	0.19				
	Cu/62.6	111	43.49	2.08	21.7	0.44	3.598			
		200	50.69	1.80	86.27	0.11				

^a^ Miller indices for corresponding planes; ^b^ spacing between planes; ^c^ average crystallite size; ^d^ full width at half maximum; ^e^ crystal lattice parameter.

**Table 4 membranes-13-00495-t004:** Parameters calculated from various kinetic models (initial Pb(II) concentration: 50 ppm, pH: 5.0).

Kinetic Model	Linearized Equation	Model Parameters	Value			
			Cu_CHOH @PET	Cu_Asc @PET	Cu_DMAB @PET	Cu_Gly @PET
Experimental data	-	$Q_{e}$, µg/g	438.5	560.0	327.0	396.0
Pseudo-first-order	$\ln\left(q_{e}-q_{t}\right)=ln\;q_{e}-k_{1}t$	$k_{1}$, min^−1^	0.01	0.009	0.009	0.009
		$q_{e}$, µg/g	926.86	726.69	252.30	511.07
		R^2^	0.88	0.89	0.98	0.90
Pseudo-second-order	$\frac{t}{q_{t}}=\frac{1}{k_{2}q_{e}^{2}}+\frac{t}{q_{e}}$	$k_{2}\times 10^{-4}$, g/µg × min	0.09	0.14	0.55	0.23
		$q_{e,}$, µg/g	588.2	666.7	357.1	454.5
		R^2^	0.99	0.99	0.99	0.99
Elovich	$q_{t}=\frac{1}{\beta }\left(\ln\alpha \beta \right)+\frac{1}{\beta }\ln t$	*α*, mg/g·min	5.89	18.28	1.24	7.85
		*β*, mg/min	0.007	0.008	0.018	0.003
		R^2^	0.99	0.98	0.94	0.99
Morris–Weber intraparticle diffusion model	$q_{t}=K_{p}t^{0.5}\;$ + C	$K_{p}$, mg/(g × h^0.5^)	14.59	72.51	10.59	13.82
		C, mg/g	197.9	487.9	100.8	76.6
		R^2^	0.97	0.92	0.76	0.97

**Table 5 membranes-13-00495-t005:** Parameters calculated from various isotherm models (initial Pb(II) concentration: 50 ppm, pH: 5.0).

Isotherm Model	Linearized Equation	Model Parameters	Sample			
			Cu_CHOH @PET	Cu_Asc @PET	Cu_DMAB @PET	Cu_Gly @PET
Langmuir	$\frac{C_{e}}{q_{e}}=\frac{C_{e}}{Q_{0}}+\frac{1}{Q_{0}b}$	$Q_{0}$, µg/g	77.52	212.57	204.08	91.74
		*b*, l/mg	52.49	979.59	1216.94	67.19
		R^2^	0.88	0.97	0.99	0.97
Freindlich	$lnq_{e}=lnk_{F}+\frac{1}{n}lnC_{e}$	$k_{F}$, mg/g	2.88	4.55	3.0	3.26
		n	1.0	1.0	1.0	1.0
		R^2^	1.0	1.0	1.0	1.0
Dubinin–Radushkevich	$lnq_{e}=lnQ_{d}-\beta \varepsilon^{2}$	β, mol^2^/kJ^2^	0.004	0.001	0.001	0.007
		$Q_{d}$, mg/g	112.6	274.3	145.7	78.3
		$E_{DR}$, kJ/mol	11.18	28.87	22.36	8.39
		*R* ^2^	0.84	0.92	0.93	0.88

**Table 6 membranes-13-00495-t006:** Comparison of Pb(II) adsorption capacity of composite TeMs loaded with copper MTs and some other sorbents.

Adsorbent	Sorption Conditions				*Q*_e_, mg/g	Ref.
	Initial Concentration of Adsorbate, ppm	Volume of Aliquote, mL	Amount of Adsorbent Utilized, mg	pH		
Chitosan/PAN composite	10.0	100.0	2000	5.0	20.08	[71]
CuO nanorods	1000.0	100.0	1000	8.5	3.31	[72]
Magnetic Fe_3_O_4_ NPs, modified with PEI	50.0	1000	1000	5.0	33.65	[73]
CuO NPs	1000	100.0	2000	6.0	37.03	[63]
ZnO NPs	10.0	25.0	20.0	6.0	22.29	[74]
Stabilized zero-valent iron NPs	50.0	10.0	100	6.0	140.8	[75]
TiO(OH)H_2_PO_4_ × 2H_2_O	10.0	300.0	1000	4.1	0.55	[76]
Cu_CHOH@PET	50.0	15.0	5.2	5.0	0.44	This study
Cu_Asc@PET	50.0	15.0	3.3	5.0	0.56	

## Data Availability

Not applicable.

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
