# Peer review of "Eco-Friendly Electroless Template Synthesis of Cu-Based Composite Track-Etched Membranes for Sorption Removal of Lead(II) Ions"

_membranes, 2023, doi:10.3390/membranes13050495_

Round 1

Reviewer 1 Report

The manuscript is well organized, and the data reported is important for the membrane technologies. Only one point in my opinion reduces the positive impression of the article. Discussion of the results of the atomic force microscopy of the obtained membranes seems to be incomplete. Also, I recommend the authors to refer some papers considering this issue:

1) Rossouw, A.; Kristavchuk, O.; Olejniczak, A.; Bode-Aluko, C.; Gorberg, B.; Nechaev, A.; Petrik, P.; Perold, W.; Apel, P. Modification of Polyethylene Terephthalate Track Etched Membranes by Planar Magnetron Sputtered Ti/TiO2 Thin Films. Thin Solid Films. 2021, 725, 138641; DOI 10.1016/j.tsf.2021.138641.

2) Golovanova, A.V.; Domnina M.A.; Arzhanov, A.I.; Karimullin, K.R.; Eremchev, I.Yu.; Naumov, A.V. AFM Characterization of Track-Etched Membranes: Pores Parameters Distribution and Disorder Factor. Appl. Sci. 2022, 12(3), 1334; DOI 10.3390/app12031334.

3) Radji, S.; Alem, H.; Demoustier-Champagne, S.; Jonas, A.M.; Cuenot, S. Investigation of Thermoresponsive Nano-Confined Polymer Brushes by AFM-Based Force Spectroscopy. Macromol. Chem. Phys. 2012, 213, 580–586; DOI 10.1002/macp.201100636.

4) Solovieva, A.B.; Timofeeva, V.A.; Erina, N.A.; Vstovsky, G.V.; Krivandin, A.V.; Shatalova, O.V.; Apel, P.; Mchedlishvili, B.V.; Timashev, S.F. Peculiarities of the Formation of Track-etched Membranes by the Data of Atomic Force Microscopy and X-ray Scattering. Colloid J. 2005, 67, 217–226; DOI 10.1007/s10595-005-0084-6.

No

Author Response

Response to the Reviewer 1 comments

We would like to thank the Reviewer for her/his interest in our work, comments and guidance that would help to improve the manuscript.

We have tried to do our best to respond to the queries raised by three reviewers and editor to clarify our research objectives and results. All the comments and questions of the reviewers were answered and necessary changes and modification in the text were made accordingly. We hope that our paper is now suitable for publication.

Query 1. The manuscript is well organized, and the data reported is important for the membrane technologies. Only one point in my opinion reduces the positive impression of the article. Discussion of the results of the atomic force microscopy of the obtained membranes seems to be incomplete. Also, I recommend the authors to refer some papers considering this issue:

1) Rossouw, A.; Kristavchuk, O.; Olejniczak, A.; Bode-Aluko, C.; Gorberg, B.; Nechaev, A.; Petrik, P.; Perold, W.; Apel, P. Modification of Polyethylene Terephthalate Track Etched Membranes by Planar Magnetron Sputtered Ti/TiO2 Thin Films. Thin Solid Films. 2021, 725, 138641; DOI 10.1016/j.tsf.2021.138641.

2) Golovanova, A.V.; Domnina M.A.; Arzhanov, A.I.; Karimullin, K.R.; Eremchev, I.Yu.; Naumov, A.V. 1.          Monárrez-Cordero, B.E.; Amézaga-Madrid, P.; Leyva-Porras, C.C.; Pizá-Ruiz, P.; Miki-Yoshida, M. Study of the Adsorption of Arsenic (III and V) by Magnetite Nanoparticles Synthetized via AACVD. Mater. Res. 2016, 19, 103–112, doi:10.1590/1980-5373-mr-2015-0667. AFM Characterization of Track-Etched Membranes: Pores Parameters Distribution and Disorder Factor. Appl. Sci. 2022, 12(3), 1334; DOI 10.3390/app12031334.

3) Radji, S.; Alem, H.; Demoustier-Champagne, S.; Jonas, A.M.; Cuenot, S. Investigation of Thermoresponsive Nano-Confined Polymer Brushes by AFM-Based Force Spectroscopy. Macromol. Chem. Phys. 2012, 213, 580–586; DOI 10.1002/macp.201100636.

4) Solovieva, A.B.; Timofeeva, V.A.; Erina, N.A.; Vstovsky, G.V.; Krivandin, A.V.; Shatalova, O.V.; Apel, P.; Mchedlishvili, B.V.; Timashev, S.F. Peculiarities of the Formation of Track-etched Membranes by the Data of Atomic Force Microscopy and X-ray Scattering. Colloid J. 2005, 67, 217–226; DOI 10.1007/s10595-005-0084-6

Response: All suggested references related to the application of AFM for membranes structure examination were cited in the revised manuscript. Also some discussions related to the AFM data were added.

Reviewer 2 Report

It is necessary to explain the practical application of the obtained membranes.

Author Response

Response to the Reviewer 2 comments

We would like to thank the Reviewer for her/his interest in our work, comments and guidance that would help to improve the manuscript.

We have tried to do our best to respond to the queries raised by three reviewers and editor to clarify our research objectives and results. All the comments and questions of the reviewers were answered and necessary changes and modification in the text were made accordingly. We hope that our paper is now suitable for publication.

Query 1. It is necessary to explain the practical application of the obtained membranes.

Response: We appreciate for this comment. Some additional explanations about practical application and advantages of prepared composite membranes were added to the introduction section of  revised manuscript (p #2)

Reviewer 3 Report

This paper reports the synthesis of composite track-etched membranes modified with electrolessly deposited copper microtubules and their testing for lead(II) ion removal capacity through batch adsorption experiments. The paper also investigates the structure and composition of the composites and determines the optimal conditions for copper electroless plating. The paper then compares the applicability of different adsorption models and shows that the Freundlich model better describes the experimental data of the composite TeMs on the adsorption of lead (II) ions. 

1. Could you please put Figure S1, Figure S2 to manuscript? 

2. Please uniform the use of "lead ions" and "Pb2+" in your paper. 

Please read the manuscript and correct small errors(such as double dot in your abstract)

Author Response

Response to the Reviewer 3 comments

We would like to thank the Reviewer for her/his interest in our work, comments and guidance that would help to improve the manuscript.

We have tried to do our best to respond to the queries raised by three reviewers and editor to clarify our research objectives and results. All the comments and questions of the reviewers were answered and necessary changes and modification in the text were made accordingly. We hope that our paper is now suitable for publication.

Query 1.  Could you please put Figure S1, Figure S2 to manuscript? 

Response: We appreciate for this comment. The main reason to place Figures S1 and S2 to the SI file is their big size. Also XRD data were presented in main manuscript as Tables #2-3. Thus we are suggest to keep figures S1 and S2 as a supplementary information.

Query 2. Please uniform the use of "lead ions" and "Pb2+" in your paper. 

 Response: corrected in revised manuscript. Instead "Pb2+” term we used equal lead(II) ions and Pb(II) ions terms.